# Flavonoid Synthesis by *Deinococcus* sp. 43 Isolated from the *Ginkgo* Rhizosphere

**DOI:** 10.3390/microorganisms11071848

**Published:** 2023-07-21

**Authors:** Jin Zhou, Kai Zou, Shaodong Fu, Zhenchun Duan, Guoqing Zhang, Xinhong Wu, Jingwen Huang, Shihui Li, Xueduan Liu, Shuangfei Zhang, Yili Liang

**Affiliations:** 1School of Minerals Processing and Bioengineering, Central South University, Changsha 410083, China; 2Key Laboratory of Biometallurgy of Ministry of Education, Central South University, Changsha 410017, China

**Keywords:** *Deinococcus*, flavonoids, genomics, LC-MS, RT-qPCR, genome-scale metabolic model

## Abstract

Flavonoids are crucial in physiological and pharmaceutical processes, especially the treatment of cancer and the prevention of cardiovascular and cerebrovascular diseases. Flavonoid-producing plants and fungi have been extensively reported, but bacteria have been much less investigated as a source of flavonoid production. *Deinococcus* sp. 43, a spherical flavonoid-producing bacteria from the *Ginkgo* rhizosphere, was reported in this study. First, the whole genome of *Deinococcus* sp. 43 was sequenced and a series of flavonoid anabolic genes were annotated. Simultaneously, High Performance Liquid Chromatography (HPLC) results showed that *Deinococcus* sp. 43 was capable of producing flavonoids, with a maximum quercetin output of 2.9 mg/L. Moreover, the relative expression of key genes involved in flavonoid synthesis was determined to test the completeness of the flavonoid anabolic pathway. The results of LC-MS analysis demonstrated that the flavonoids produced by *Deinococcus* sp. 43 were significantly different between intracellular and extracellular environments. The concentration of multiple glycosylated flavonoids was substantially higher in extracellular than intracellular environments, while the majority of flavonoids obtained in intracellular environments were hydroxylated multiple times. Lastly, the flavonoid biosynthetic pathway of *Deinococcus* sp. 43 was constructed based on the genomic analysis and the detected flavonoids. In conclusion, this study represents the first comprehensive characterization of the flavonoid-producing pathway of *Deinococcus*. The findings demonstrate that the strain has excellent potential as a genetically engineered strain for the industrial production of flavonoids.

## 1. Introduction

As a drug with great commercial value, flavonoids are synthesized by the phenyl-propyl pathway in plants, bacteria, and fungi [1]. They are universally present in a variety of plants. They have the basic structure of C6-C3-C6, which is formed by connecting the A ring and B ring through a C ring containing three carbon atoms [2]. So far, over 10,000 kinds of flavonoids have been identified. Specific modifications of molecules caused by hydroxylation, methylation, glycosylation, or acylation of A and B rings form different classes of flavonoid compounds. A large number of studies have revealed that flavonoids have physiological and pharmacological activities in human health, such as antioxidant, anti-cancer, and prevention of cardiovascular and cerebrovascular diseases [3,4,5,6]. Therefore, flavonoids are widely added to drugs and health products. It has been reported that grape skins, an industrial by-product, are rich in proanthocyanidins, so the addition of grape skins to animal feed is promising to replace traditional antibiotics [7]. In recent years, people’s demand for flavonoids has been increasing. The majority of flavonoids on the market are derived from plant extracts, such as *Celery*, *Ginkgo*, and *Soybean*. Yet, environmental variables strongly influence plant growth, and the sources of flavonoids extracted from plants are limited. These facts indicated that plant extraction cannot yet meet the market demand for flavonoids [8,9,10].

The flavonoid synthesis pathway includes many essential enzymes, such as CHS (chalcone synthase) and CHI (chalcone isomerase). CHS and CHI catalyze the formation of necessary precursors for flavonoid synthesis. Malonyl-CoA and phenylalanine were used as precursors to synthesize chalcone under the catalysis of PAL (phenylalanine ammonialyase), C4H (cinnamate-4-hydroxylase), CHS (chalcone synthase), and other enzymes. The chalcone was catalyzed by CHI to synthesize naringin, which is a key intermediate in flavonoid production. Then, a group of downstream flavonoid synthases catalyzes the precursor to generate a wide range of flavonoids. F3H (flavanone 3-hydroxylase) and FLS (flavonol synthase) are required for the synthesis of flavonols and their derivatives. The synthesis of anthocyanins requires the catalysis of DFR (dihydroflavonol 4-reductase), ANS (anthocyanin synthase), and FNS (flavonoid synthase), which synthesize flavonoids and their derivatives. In addition, a large number of methylation, hydroxylation, and glycosylation catalytic enzymes have been found. The presence of multiple enzymes promotes the diversity of flavonoids synthesized [11].

The plants are inhabited by many different microbial communities that provide fitness benefits to their respective hosts [12,13]. Since plants have a long-term co-evolutionary relationship with endophytic microorganisms, this may allow many endophytes to obtain enzyme genes from certain metabolic pathways of the host during evolution [14,15,16]. Microorganisms acquiring these genes may have the ability to directly or indirectly biosynthesize metabolites, which perform specific functions in the host. Plant development, environmental adaptation, and even the evolution of plant–microbe interactions are all significantly influenced by these dynamic interactions. Because of this, different plant endophytes have been isolated and found to produce natural products, which include paclitaxel, kombucha, vinblastine, podophyllotoxin, isoindoline-1-one, taraminolactone (triamcinolone), cellular acids, alternating silanol, cryptosentin (cryptoxanthin), and rutin [17,18,19,20,21]. These findings provide a new direction for the industrial production of flavonoids using microorganisms.

In recent years, through the isolation and identification of endophytic microorganisms in different tissues of *Ginkgo*, a large number of flavonoid-producing microorganisms have been found, most of which are *Aspergillus* and *Actinomycetes* [22,23,24,25]. Moreover, some microorganisms also have the ability to utilize or convert flavonoids. Nevertheless, most of the microorganisms that produce natural active substances are fungi. There are various issues with fungus fermentation, such as the lengthy period and difficulty in extracting the products. The use of bacterial fermentation, however, can lessen these issues.

*Deinococcus* is a type of bacteria that is highly resistant to radiation and can survive and reproduce in extreme environments, which is considered to have important application prospects. Some researchers have found that *Deinococcus* has the function of flavonoid transformation. The substrate quercetin is converted to quercetin 4′-O-d-isomaltoside by starch sucrase catalysis in *Deinococcus* [26]. Moreover, it has been reported that the growth of *Deinococcus* was better than that of the untreated one when flavonoids were added to the R2A medium [27]. 

In this study, *Deinococcus* sp. 43 was isolated from the natural *Ginkgo* rhizosphere. Essential genes involved in flavonoid synthesis of the strain *Deinococcus* sp. 43 were identified through real-time quantitative polymerase chain reaction (RT-qPCR), which helped construct the flavonoid biosynthesis pathway. Based on genomic analysis and metabolite identification, the flavonoid biosynthesis pathway of *Deinococcus* sp. 43 was reconstructed. Furthermore, through the precision detection equipment, the flavonoid production was measured at different fermentative conditions. The study suggested that this bacterium *Deinococcus* sp. 43 has great potential applications for flavonoid production.

## 2. Materials and Methods

### 2.1. Bacterial Genome Extraction

An endophytic bacterium was isolated from the root tissue of *Ginkgo biloba*, which grew in Linyi City, Shandong Province, China (34°36′34″ N, 118°12′8″ E). It was cultured on Reasoner’s 2A Medium Agar (R2A) at 25 °C for 7–10 days. The genomic DNA was extracted by sodium dodecylsulfate (SDS). For DNA extraction, about 1.5 mL of bacterial solution was used to extract the total genomic DNA of the bacteria according to the manufacturer’s protocol and recommendations. A NanoDrop ND-83 1000 spectrophotometer (NanoDrop Technologies, NC, USA) was used to determine the purity and concentration of DNA. Stained TAE-agarose gel and Elution Buffer (EB) were used to detect the extracted DNA’s quality. The extracted DNA samples were subjected to 16S rDNA polymerase chain reaction (PCR); the primer sequences were 27F (5′-AGAGTT TGATCCTCGCTCAG-3′) and 1492R (5′-TACGGYTACCTTGTTACGACTT-3′). The sequence obtained was then submitted to BLASTN for sequence homology approximation. Neighbor-joining (NJ) trees were applied for the phylogenetic analysis using the Molecular Evolutionary Genetics Analysis (MEGA) X software. After that, the filtered strains were used for the subsequent genome sequencing and bioinformatics analysis.

### 2.2. Genome Sequencing, Assembly and Functional Annotation for Flavonoid Synthesis

The sequencing work was conducted at the Beijing Genome Institute. The size of the DNA fragments was about 200~400 bp, and these were sequenced on the BGISEQ-500 sequencing platform. For a sufficient sequencing depth for assembly, approximately 1Gb of data was produced for the bacteria. Clean data were yielded by removing ploy-N, and low-quality reads from raw data. Clean reads from the sequencing of the bacterial genome were assembled by SPAdes (v3.14.0) [28]. The results of the bacterial genome assembly were evaluated using CheckM [29]. Meanwhile, genome annotation was performed using Prokka (v1.14.6) [30]. Functional annotation was finished via BLAST with different public databases. Blasting against the NR database was performed using the diamond (v2.0.4) software [31]. KAAS system was used for annotation and pathway mapping [32] and the e-value was 1 × 10^−6^. The COG classification statistics of all the predicted amino acid sequences were predicted and analyzed in the eggnog-mapper [33,34]. Then, the sequence with the highest similarity was selected and the evolutionary tree was constructed using MEGA X. Subsequently, the metabolic pathway was reconstructed in iPath3 [35].

### 2.3. HPLC Sample Preparation and Detection Conditions

First, 200 mL of sterile TGY liquid medium was set as the blank control group. All liquid media were placed in a shaking incubator at 180 rpm at 28 °C for 60 h. After cultivation under specific conditions, the fermentation broth was directly sonically broken and extracted. The crushed fermentation broth was then mixed with ethyl acetate for 12 h for extraction. The upper extract was collected (ethyl acetate). Then, it was concentrated using a rotary evaporator. After the organic phase was evaporated, we rinsed the walls of the bottle with 5 mL of methanol and collected the extract. Lastly, it was filtered using a 0.22-micron polarity filter membrane. 

The detection conditions for HPLC were set as follows: autosampler set to 10 μL injection volume for a total flow rate of 1.0 mL/min (A:B = 1:1); pure acetonitrile (A), double distilled water (B); the column temperature was 35 °C; the detector was set to a detection wavelength of 360 nm; the extraction elution procedure was set to 0–10 min to stop; column: YMC-TriartC18 (5 μm, 250 mm × 4.6 mm). Meanwhile, we accurately weighed 10 mg of quercetin, 20 mg of kaempferol, and 10 mg of isorhamnetin, and then the three standards were mixed. The standard stock solution was obtained by using 25 mL of methanol at a constant volume. The standard stock solution was diluted to 2×, 5×, 10×, 25×, and 50× to prepare the standard curve.

### 2.4. LC-MS/MS Sample Preparation and Detection Conditions

For LC-MS samples, after cultivation, centrifugation of cell fermentation broth and collection of the supernatant to detect extracellular flavonoids were carried out, and precipitated bacterium were used to measure intracellular flavonoids. The extraction process was similar to that of HPLC; the difference is that the precipitated bacterium needed to be cleaned with distilled water.

The detection conditions for LC-MS/MS were set as follows: analysis was carried out using the Shim-pack UFLC SHIMADZU Nexera X2 liquid chromatographic system. LC separations were accomplished using an Agilent SB-C18 column (1.8 µm, 2.1 mm × 100 mm) at 40 °C. A flow rate of 0.35 mL/min was chosen and 0.1% acetic acid water (A) and 0.1% acetic acid containing acetonitrile (B) comprised the mobile phase. The gradient elution program was as follows: 0~9.0 min, 5→95% B; 10.0~11.0 min, 95→95% B; 11.0~11.1 min, 95→5% B; 11.1~14.0 min, 5→5% B. The injection volume was 4 μL. MS was carried out using the Applied Biosystems 4500 QTRAP system (AB SCIEX Technologies, Road Redwood City, CA, USA), equipped with an electrospray ionization (ESI) and UHPLC system to scan 100 to 1500 molecular weight parent ions at 550 °C. Other MS parameters were as follows: MS voltage, 5500 V; curtain gas (CUR), 25 psi; collision-activated dissociation (CAD), high; declustering potential (DP) and collision energy (CE) and specific optimization.

### 2.5. Metabolite Data Processing and Analysis

The acquired MS raw data were baseline filtered and peak identification, integration, retention time correction, peak alignment and normalization were performed using the software Analyst 1.6.3 (AB SCIEX Technologies, Road Redwood City, CA, USA), including peak recognition, alignment, calibration of the internal standard, filtering and normalization to the total area. Analyst 1.6.1 (AB SCIEX Technologies, Road Redwood City, CA, USA) was recommended to visualize raw data of target components in two-stage mass-to-charge ratio maps. Based on the MWDB database (Wuhan Metware Biotechnology Co., Ltd., Wuhan, China) and metabolite information public database, the primary and secondary spectrum data of mass spectrometry were analyzed qualitatively. The structure of the metabolites was analyzed with reference to MassBank (http://www.massbank.jp/, accessed on 31 May 2022), KNAPSAcK (http://kanaya.naist.jp/KNApSAcK/, accessed on 31 May 2022), HMDB (http://www.hmdb.ca/, accessed on 31 may 2022), MoToDB (http://www.ab.wur.nl/moto/, accessed on 31 May 2022), METLIN (https://metlin.scripps.edu/index.php, accessed on 31 May 2022) and other MS public databases.

### 2.6. Gene Expression Analysis by RT-qPCR

Total RNA was extracted with Trizol Reagent (Sangon Biotech, Shanghai, China) according to the protocol of the manufacturer. RNA quality was examined by electrophoresis on a 2.0% agarose gel, and its amount was analyzed by a spectrophotometer (NanoDrop 2000, Thermo Scientific, Waltham, MA, USA). For RT-qPCR analysis, cDNA was synthesized with the Moloney Murine Leukemia Virus Reverse Transcriptase (Sangon Biotech, Shanghai, China) according to the protocol of the manufacturer. The reaction volume was set to 20 μL in accordance with the operation manual of the 2xSG Fast qPCR Master Mix (Sangon Biotech, Shanghai, China). DNA was amplified with a real-time PCR system (QuantStudio 1, Applied Biosystems by Thermo Fisher Scientific, Bend, OR, USA) to analyze the expression of target genes under the following PCR procedure: denaturation of 3 min at 95 °C and 30 cycles of 95 °C for 10 s and 55 °C for 30 s and 72 °C for 60 s. The primers used for amplification of the reference gene RLPL32 and target genes are presented in Table 1.

## 3. Results

### 3.1. Strain Identification

A bacterium named *Deinococcus* sp. 43 was isolated from the *Ginkgo* rhizosphere. GenBank’s BLAST program was used to calculate the 16S rRNA gene sequence similarity between *Deinococcus* sp. 43 and the related bacteria. The sequence of 16S rDNA exhibited maximum similarity (98.72%) with the *Deinococcus soli* (exCha et al., 2016) strain N5. The phylogenetic consensus tree was built from eight sequences of the 16S rRNA (seven references and one clone) by the neighbor-joining (NJ) method using MEGA X software (Appendix A). Combined with phylogenetic analysis and sequence homology, strain 43 was named *Deinococcus* sp. 43.

### 3.2. Genome Sequencing, Assembly and Functional Annotation for Flavonoid Synthesis

A total of 4,017,618 clean reads (BGISEQ-500, 2 × PE150) were obtained to include 1,205,285,400 clean bases for *Deinococcus* sp. 43, and its Q20 was 96.28%. After spade stitching of read fragments and Quast software 5.1.0. evaluation, *Deinococcus* sp. 43 had a genome of 4,314,008 bp with a 203,832 bp N50, and the G + C content was 69.53%. The bacterium’s overall annotation rate was 99.57%, indicating the high quality of genome assembly. *Deinococcus* sp. 43 contains 4116 CDSs, 56 tRNA, 2 rRNA genes, and 1 tmRNA sequence, respectively. Of these, 4036 sequences (96.67%) were annotated in the NR database, 3252 CDSs (77.89%) in the COG database, and 1808 CDSs (43.31%) in the KEGG database. The annotation results show that the NR database has the highest annotation success rate (Table 2). By comparison with the COG database, the annotated genes were mapped to 28 functional taxa. It has a high proportion of genes mapped to functional groups called the regulation of biological processes, metabolic processes, catalytic activity, and cellular processes. Among them, COG functional annotation classification results show that 13.6% of the genes were mapped to “aging metabolism” (6.8%) and “cellular process” (6.8) in Figure 1.

Flavonoids have a particular ability to resist oxidative damage. *Deinococcus* sp. 43 uses the complete pathway of flavonoid synthesis to form the basic C6-C3-C6 flavonoid monomer. By comparing the NR and COG database databases, the coding genes corresponding to PAL, 4CL and CHS were annotated. We built a flavonoid synthetic protein library by ourselves, used diamond (v2.0.4) software to compare this with the genome of *Deinococcus* sp. 43, and a constructed phylogenetic tree. We found that several proteins have high similarity with key flavonoid synthesis enzymes, such as CHI, C4H, and 4CL; these enzymes are at the beginning of the flavonoid synthesis pathway. Other enzymes including F3H, FLS, and F3′M in major metabolic pathways were also identified, as key catalytic enzymes; these enzymes play an important role in the further catalysis of flavonoid substances. We also found several genes involved in flavonol metabolism, such as genes encoding CYP75A (flavonoid 3′,5′-hydroxylase), R06611 (flavonol 3-O-glucosyltransferase) and flavonoid 3′-o-methyltransferase. These enzymes can further modify quercetin or kaempferol to produce other types of flavonols. Many genes that are downstream have also been discovered. Several amino acid sequences share a high degree of similarity with the enzymes that catalyze the synthesis of anthocyanins, including ANS, DFR, BZ1 and UGT75C1, suggesting that *Deinococcus* sp. 43 may have the ability to produce anthocyanins. It is worth noting that most of the proteins that catalyze anthocyanin synthesis have been found in plants, and a few in bacteria or fungi. Based on these findings, we hypothesized that these genes in *Deinococcus* sp. 43 were acquired during frequent HGT, thus acquiring the ability to synthesize flavonoids (the results and analysis are presented in the file of Appendix A). 

### 3.3. Quercetin Production Analysis and Optimization

Considering the presence of key genes for flavonoid synthesis, we carried out HPLC detection of fermentation broths. HPLC chromatograms of bacterial fermentation broths and standards are described in Appendix A. The yield of flavonoids from the bacterium can be calculated by comparing the peak area with the standard sample (c). As can be seen from Appendix A, the content of quercetin is about 1.6 mg/L in 35 g/L TGY medium, which is significantly different from the blank medium, which means the bacterium has quercetin-producing ability. However, kaempferol and isorhamnetin were almost absent. 

The level of quercetin represents the quantity of carbon flux along the flavonoid metabolic pathway. Therefore, quercetin content was used to measure flavonoid dynamics. We can observe that the quercetin yield increased with the increasing medium concentration; the highest quercetin yield (1.9 mg/L) was obtained in the medium of 45 g/L (Figure 2)**.** In a 10 g/L TGY medium, *Deinococcus* sp. 43 was observed to produce minimal quercetin; 1.6 mg/L quercetin was produced by *Deinococcus* sp. 43 in a 35 g/L TGY medium. The regular variation between the medium concentration and quercetin yield suggests that additional sources of carbon are needed for quercetin production. In the gene expression experiment, we found that the expression level of flavonoid synthesis genes with the 45 g/L medium concentration was dozens of times higher than that of 10 g/L.

Due to the complex synthetic and regulatory mechanisms of *Deinococcus* sp. 43, the effect of fermentation time on the variation in flavonoid production was investigated.

The quercetin production of *Deinococcus* sp. 43 generally showed a trend of first rising and then decreasing. Within 0–24 h, quercetin is not produced. At 72 h, quercetin production reached its peak. The growth rate of quercetin is maximal between 48 and 72 h, and the quercetin yield of *Deinococcus* sp. 43 at 72 h (2.9 mg/L) was 5.0 times higher than that at 48 h (0.6 mg/L). The results of RT-qPCR showed that the expression of key genes for flavonoid synthesis at 72 h was about 2 times higher than that at 24 h.

After 72 h, the content of quercetin began to decrease rapidly, which may be the result of the transformation or utilization of quercetin by the bacteria because we had genes that can convert quercetin to myricetin (flavonoid 3′,5′-hydroxylase). As the biomass reaches its peak (48 h), quercetin levels start to rise quickly, indicating that quercetin accumulation started during the stationary phase and ended during the decline phase. These findings demonstrate that the production of quercetin is tightly related to the growth phase, particularly around the stationary phase.

### 3.4. Analysis of Flavonoid Metabolites

It should be noted that 76 flavonoid types have been identified based on the local database of flavonoids, including 38 flavones, 17 flavonols, 6 flavanones, 2 isoflavones, 4 anthocyanidins, and 6 proanthocyanidins (the mass spectra are shown in the Appendix A). After removing the blank control signal, there were 56 kinds of metabolites in the sample. Since flavonoids are also present in the medium, it is of great importance to know whether the synthesis of flavonoids by *Deinococcus* sp. 43 is de novo or through intermediates present in the medium. Naringin is an important precursor to the production of flavonoids. Through the analysis of naringenin content, cells secrete more naringenin than naringenin contained in the medium, so we conclude that the bacteria has the ability to synthesize naringenin by itself, that is, the ability to synthesize flavonoids de novo. After removing the flavonoids already present in the original medium, Table 3 shows the 56 different types of flavonoids that were found in the bacterium. It also demonstrates the intensity of the flavonoid signal in *Deinococcus* sp. 43 intracellular and extracellular environments. Depending on the variation in total signal intensity, 3,4′-dihydroxyflavone, tricin-4′-O-(guaiacylglycerol) ether, tricin-7-O-rutinoside, 5,7,2′-trhiyroxy-8-methoxyflavone, tricin (5,7,4′-Trihydroxy-3′,5′-dime thoxyflavone), apigenin-6-C-glucoside (isovitexin), and naringenin chalcone had higher signal intensity.

The metabolome results also showed that the relative concentrations of the flavonoids varied considerably in and out of the cell, as presented in Figure 3. From the logarithm of the ratio of signal intensity of metabolite ions (log2FC), Figure 3 depicts 20 metabolites with the most diverse intracellular and extracellular concentrations. Intracellular substances such as tricin-4′-O- (guaiacylglycerol) ether, 5,7,2′-trhiyroxy-8-methoxyflavone, tricin, tricin-7-O-neohesperidoside, apigenin-6-C-glucoside (isovitexin), naringenin chalcone, and chrysoeriol are comparatively higher than the extracellular content. Meanwhile, tricin-7-O-(2′-O-rhamnosyl), chrysoidium-6,8-di-C-glucoside, and cyanidin-3-O-arabinoside accumulate predominantly in extracellular environments. As a result, hydroxylated and methylated flavonoids accumulate more intracellularly, while glycosylated flavonoids accumulate more extracellularly. Not only is this evidence that *Deinococcus* sp. 43 has enzymes that catalyze the formation of these flavonoids, but it should also be investigated why this intracellular and extracellular difference occurs.

The above figure depicts 20 metabolites with the most diverse intracellular and extracellular concentrations. The red part is a much more extracellular substance than intracellular substance, whereas the green part is much less intracellular than extracellular.

### 3.5. Construction of the Genome-Scale Metabolic Model of Deinococcus sp. 43

The construction of a genome-scale metabolic model is an efficient method for understanding the overall metabolic network and predicting the potential applications of microorganisms for producing valuable metabolites. Distant species sharing similar secondary metabolic pathways are a very common phenomenon in nature. Therefore, according to the elucidated flavonoid metabolic pathway, combined with the flavonoids measured by LC-MS and the genes associated with flavonoid synthesis and derivation, we constructed the metabolic pathway of *Deinococcus* sp. 43, as shown in Figure 4.

Similar to the metabolic model revealed in previous studies, when *PAL* and *TAL* were compared at the same time, we determined that *Deinococcus* sp. 43 follows the order of synthesis from phenylalanine or tyrosine to flavones, flavonols, anthocyanidins, and some other flavonoids. It demonstrates almost complete upstream flavonoid synthesis to form the basic C6-C3-C6 flavonoid monomer. Flavones, flavonol, and proanthocyanidin are all detected in metabolites. At the same time, in the metabolites of *Deinococcus* sp. 43, the presence of anthocyanin substances is found, which reveals two cases: 1. the bacterium has a complete pathway for the production of anthocyanin and has a complete flavonoid synthetase enzyme system; 2. the bacterium can use some anthocyanin synthesis precursors in the medium to produce anthocyanins, which means that *Deinococcus* sp. 43 has some enzymes that can catalyze the synthesis of anthocyanins. At present, there is still a large gap in the anthocyanin market, and most of the anthocyanins come from superplant extraction. These may provide an important reference for the industrial fermentation and production of anthocyanins. Because it has a complete flavonoid synthetase system, in the meantime, it can be used as a chassis bacterium to greatly simplify the operation of genetic engineering.

## 4. Discussion

### 4.1. Deinococcus Becomes a Potential Source for Flavonoid Production

Plant endophytes are a source of large quantities of bioactive metabolites [36]. As a result of gene transfer, endophytes may acquire certain metabolic genes of host plants during long-term co-evolution with plants, so many endophytes have acquired the same ability to produce natural products as their hosts during co-evolution. Zou found that genes of certain functional enzymes key to the flavonoid metabolic pathway in *Streptomyces* sp. Gbtc1 resemble those of the ginkgo host [37]. The endophytes that produce paclitaxel, for instance, have been found in numerous studies [38,39]. Moreover, numerous plant endophytes such as actinomycetes and fungi that produce flavonoids have been discovered. Even though many organisms that produce flavonoids have been isolated, not much is known about the synthetic pathway by which these microorganisms produce flavonoids. Nevertheless, it is uncommon that the majority of these isolated metabolite-producing endophytes are bacteria. Due to the high medical usefulness of flavonoids, numerous genetically engineered microbes that produce flavonoids have been developed. However, there are not enough genes for some bacterial adaptations, which means many kinds of important flavonoids cannot be obtained by engineered bacteria. It is infinitely important to identify enzyme genes derived from microorganisms to solve this issue. At the same time, the flavonoid production of these endophytic microorganisms was not high. The first bacterium known as *Deinococcus* sp. 43 has been found to create flavonoids from scratch using carbon sources and we reveal the flavonoid anabolic pathway of *Deinococcus* for the first time. This is an addition to the flavonoid-producing microbes that are currently known.

The major genes involved in the known flavonoid production upstream pathways include *PAL*, *CHS*, *CHI*, *F3H*, etc. At present, most of the upstream genes for flavonoid anabolism are found in microorganisms. However, many genes in the downstream synthetic pathway of flavonoids have not been reported, and many downstream flavonoids are of great value. In this study, we found that *Deinococcus* contains a large number of genes downstream of the flavonoid synthesis pathway, in particular, anthocyanin synthesis genes. At the same time, flavonoid-producing microorganisms have great prospects for the industrial production of flavonoids as chassis bacteria due to their genes related to flavonoid synthesis. Based on the construction of metabolic pathways, researchers can refer to them and perform genetic operations to obtain their desired products. At present, no studies have reported anthocyanin-producing microorganisms. *Deinococcus* can be a strain for achieving large-scale industrial anthocyanin production or may offer suitable genes for the creation of genetically modified bacteria that produce anthocyanins.

### 4.2. The Intricate Mechanism of Deinococcus’ Production of Flavonoids

*Deinococcus* is known for its extreme radiation resistance, but its anti-radiation mechanism is still being explored. Many studies have illustrated that the active substance is of great importance for resistance. Daly’s study shows that the degree of bacterial resistance is determined by the level of oxidative protein damage [40]. From this point, researchers have also found active antioxidant substances in *Deinococcus*, such as deinoxanthin, alloxanthin, and lycoxanthin [41], These types of carotenoids can react with free radicals via their conjugated double bonds to block other free radical reactions [42]. Numerous studies have shown that flavonoids can scavenge free radicals within cells. In our study, we found that *Deinococcus* sp. 43 produces a variety of flavonoids; we speculate that flavonoids act in synergy with carotenoids to control free radicals in *Deinococcus* sp. 43 under pressure to maintain stability. 

In addition, relevant studies have proved that high concentrations of flavonoids in plant roots are conducive to the colonization of host plants by microorganisms [43]. Since *Deinococcus* sp. 43 was isolated from the root of the *Ginkgo*, we assume that another reason for bacteria to secrete flavonoids is to better colonize the host and inhibit other bacterial pathogens. *Ginkgo* is one of the most radiation-resistant organisms; this is why the species survived in the Paleozoic era 345 million years ago (a highly radioactive environment). To a certain extent, the anti-radiation and repair mechanisms of Ginkgo biloba are similar to those of *Deinococcus*; perhaps this involves gene transfer in which the two organisms live in symbiosis for a long time. At the same time, many plants need to produce flavonoids against radiation in the environment; some endophytes can help hosts produce flavonoids to resist ultraviolet radiation [44,45], or produce flavonoids for the host directly [46]. *Deinococcus* sp. 43 may play an important role within the *Ginkgo* host and could be another important reason why *Ginkgo* survives in high-radiation environments. 

When the medium concentration is less than 20 g/L, the bacteria do not produce quercetin. The mechanisms involved in this can be multifaceted. From a macroscopic perspective, the higher the nutrient content, the more primary metabolites are produced and the more raw materials are used to synthesize secondary metabolites. Microscopically, studies have shown that the type, quantity, and intake of nutrients also affect gene expression in some metabolic pathways of bacteria to some extent [47,48]. In *Arabidopsis,* the frequency of certain genetic expressions of flavonoid metabolic pathways increases with the consumption of nitrogen and phosphorus, as does the level of flavonoids [49]. We identified differences in the expression of key genes by RT-qPCR. Fermentation time also had great influence on quercetin yield, especially after 72 h of incubation. The rapid decline in quercetin content may indicate a special function of the bacterium, which is consistent with Farzaneh Sadat Naghavi’s findings. In *Rhodotorula slooffiae* and *R. mucilaginosa* isolated from leather tanning wastewater, carotenoids also have a large reduction after reaching their maximum yield after 48 h [50]. There is the possibility of quercetin being metabolized into another essential substance as an intermediate. There are several experimental phenomena to support this hypothesis. *Amylosucrase in Deinococcus geothermalis* can perform a series of glycosylated modifications to quercetin [26]. Steed found that the gut microbiome-impaired clostridial anaerobe can degrade flavonoids, producing deaminotyrosine (DAT), which promotes interferon release [51]. Westlake. D.W.S. found that *Aspergillus flavus* degrades rutin into protocatechuic acid, resorcinol carboxylic acid, rutin sugar, and carbon monoxide, and discovered quercitrin was used by *Pullularia*, *Cryptococcus albidus*, and *Cryptococcus difluens* [52,53]. *Clostridium orbiscindens* isolated from human feces has been proven to convert quercetin and taxifolin to 3,4-dihydroxyphenylacetic acid [54].

### 4.3. The Variations in the Intracellular and Extracellular Distribution of Flavonoids Are Correlated with Their Distinct Biological Roles

Flavonoids, which are produced as a result of plant secondary metabolism, are a diverse class of polyphenolic compounds involved in UV scavenging, fertility, and disease resistance and play a multifunctional role in plant–microbe and plant–plant communication [55,56,57,58]. The diversity of flavonoid structures is due to the basic skeletal changes through enzymes such as glycosyl transferases, methyl transferases, and acyl transferases [59]; structural differences often result in large functional differences. In our experiment, the types and concentrations of intracellular and extracellular flavonoid metabolites in *Deinococcus* sp. 43 were different to a certain degree. Flavonoid metabolites accumulate significantly in the cell and are modified by hydroxyl groups, whereas flavonoids congregate in large quantities outside the cell and are mostly modified by glycosyl groups. The apparent reason is that different types of flavonoids have different secretory capacities. Currently, there are few studies on the reasons why flavonoids show differences in intracellular and extracellular distribution, but there are many studies on changes in flavonoids.

Flavonoid modifications include methylation, glycosylation, acylation, hydroxylation, and so on. The locations as well as the quantity of modification groups have a significant impact on the biological activities of flavonoids [57,58,59]. According to Shi’s study [56], methylation at positions 4 and 7 has a strong inhibitory effect on cancer cell lines, but methylation at position 5 is less obvious. According to Chen [60], glycosylation increases the chemical stability and solubility of luteolin. Moreover, Alseekh discovered that hydroxylation and acylation can enhance the antioxidant activity of flavonoids to a limited extent [10]. In addition, related studies have revealed that the more hydroxyl groups on the B-ring, the greater the free radical scavenging activity [61]; and possibly, hydroxylation plays a role in maintaining intracellular free radicals at a normal level [10,62,63,64]. When *Deinococcus* sp. 43 is exposed to intense radiation or other coercive environments, the bacteria not only need to repair the broken double-strand but also quickly remove the high concentration of free radicals brought by radiation to avoid the devastating impact of the bacteria. Our findings show that intracellular flavonoids are mostly hydroxylated modifications, and we speculate they can quickly help remove harmful substances within the cell. There are also some other biological roles involved; studies have shown that flavonoids can bind to proteins to cause quenching, and hydroxylated flavonoids have a significantly stronger binding capacity than glycosylated flavonoids [65]. Flavonoids are secondary metabolites that occur late in growth, and high intracellular concentrations of hydroxylated flavonoids may be associated with decreased cell metabolic capacity during specific periods, allowing bacteria to use nutrients.

As extracellular flavonoids are primarily glycosylated modifications, the first finding is that *Deinococcus* sp. 43 contains a sizable number of enzymes that catalyze flavonoid glycosylation; this has also been noted in other members of the same genus, “*Deinococcus geothermalis*” [66]. They may also be crucially connected to specific bacterial functions. According to reports, amino acids are *Deinococcus’* preferred carbon source, and glucose is only utilized when those sources are no longer available [67]. In the presence of sufficient carbon sources, glucose and flavonoids may combine to generate glycosylated flavonoids, which have significantly better solubility, storage resistance, and stability. Nevertheless, the antioxidant capacity decreased as a result of flavonoid glycosylation [68,69,70]. We hypothesized that this might be a means for bacteria to reserve antioxidants and carbon sources for future use. Flavonoids have been demonstrated in recent years to function as quorum-sensing inhibitors. Flavonoids have potency in the range of 3.69 to 23.35 M in *Chromobacterium violaceum*, where they block the transmission of acyl allo-glucosalolactone (AHL)-type QS signaling [71]. The chrysin derivative attaches to the bacterial QS regulator LasR in vitro tests, which renders it incapable of binding to DNA [72]. In this experiment, there are a lot of glycosylated flavonoids that are present outside of the cell. We hypothesize that these flavonoids may be involved in the transduction of quorum sensing signals, possibly by quenching some specific proteins or by interacting synergistically with intracellular hydroxylated flavonoids that are involved in protein quenching, to allow the bacterial population to survive in a dormant state.

## 5. Conclusions

In this study, the flavonoid synthetic pathway was reconstructed in the bacterium *Deinococcus* sp. 43 isolated from the *Ginkgo* rhizosphere for the first time. Based on genomic and metabolite analysis, the key functional enzymes expressed by these genes constitute an almost complete pathway of flavonoid synthesis, which indicated that *Deinococcus* sp. 43 has the potential to synthesize flavonoids. The medium concentration was shown to have an effect on the synthesis of flavonoids by modifying the expression of crucial genes as well as quercetin’s dynamics with fermentation time. There are significant differences between the types of intracellular and extracellular flavonoids; hydroxylated flavonoids mostly aggregate intracellularly, while extracellular flavonoids are essentially glycosylated flavonoids. The causes of this phenomenon need to be further explored. These findings provide useful supplemental information about *Deinococcus*. The strain in particular has excellent application potential as a genetically engineered strain for the industrial production of flavonoids, and the results of this research provided additional information on the production of flavonoids by bacteria.

## Figures and Tables

**Figure 1 microorganisms-11-01848-f001:**
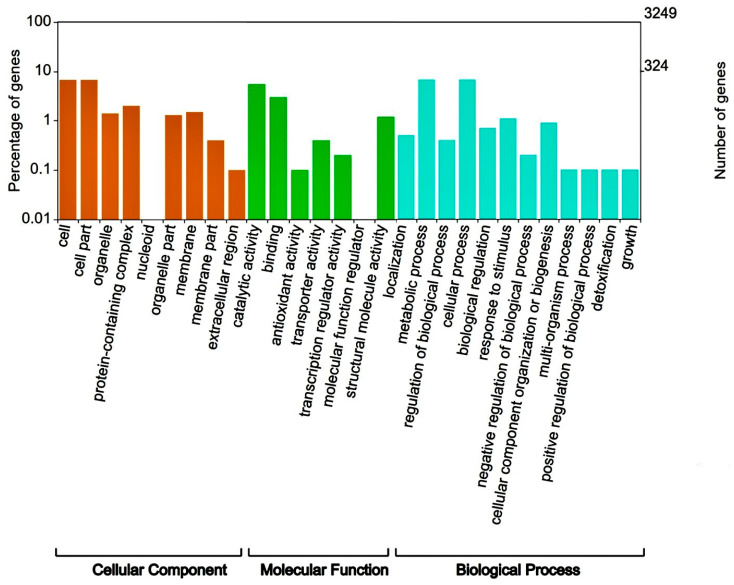
COG functional annotation classification of *Deinococcus* sp. 43.

**Figure 2 microorganisms-11-01848-f002:**
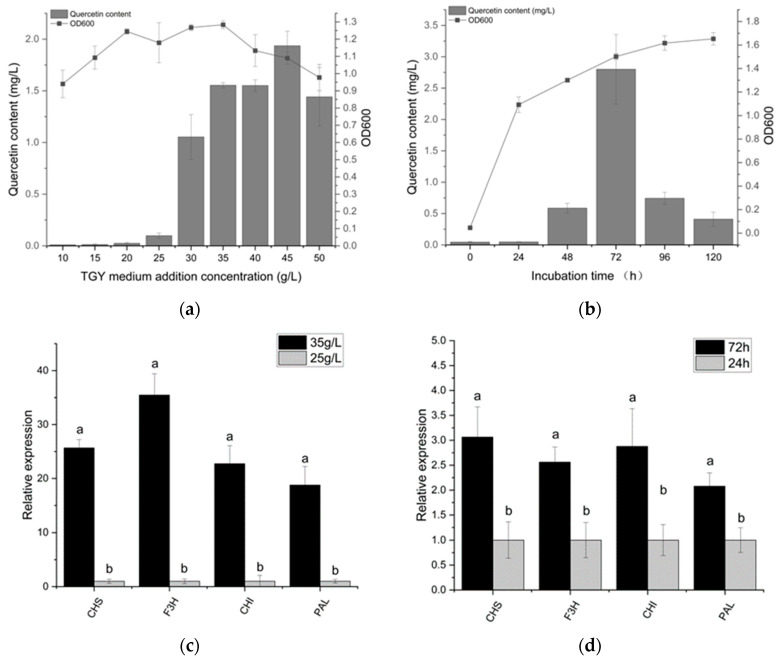
It shows the growth of bacteria and quercetin production at different TGY concentrations and fermentation times. (**a**) demonstrates how the concentration of the medium has a significant impact on the yield of quercetin. In the medium containing 45 g/L, the highest quercetin yield (1.9 mg/L) was attained. When the medium concentration was less than 25 g/L, a small amount of quercetin was produced by *Deinococcus* sp. 43. On the whole, the yield of quercetin is positively correlated with the medium concentration. (**b**) shows the change in quercetin in fermentation broth at different times. The quercetin production of *Deinococcus* sp. 43 generally showed a trend of first rising and then decreasing. Quercetin production reached its maximum at 72 h. In response to variations in flavonoid production, both (**c**,**d**) demonstrate the relative expression of some essential genes involved in flavonoid synthesis. We can find that the expression of flavonoid synthesis genes increases with increased flavonoid production. In particular, under different medium concentrations, the expression of flavonoid synthesis genes varied tens of times. The different letters in the figure indicate significant differences.

**Figure 3 microorganisms-11-01848-f003:**
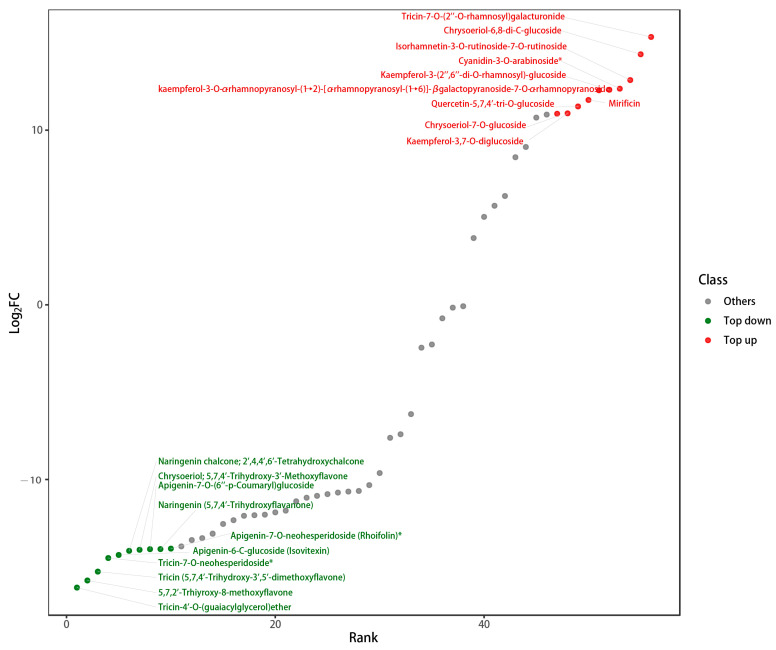
Intracell and extracellular differences of flavonoid substance.

**Figure 4 microorganisms-11-01848-f004:**
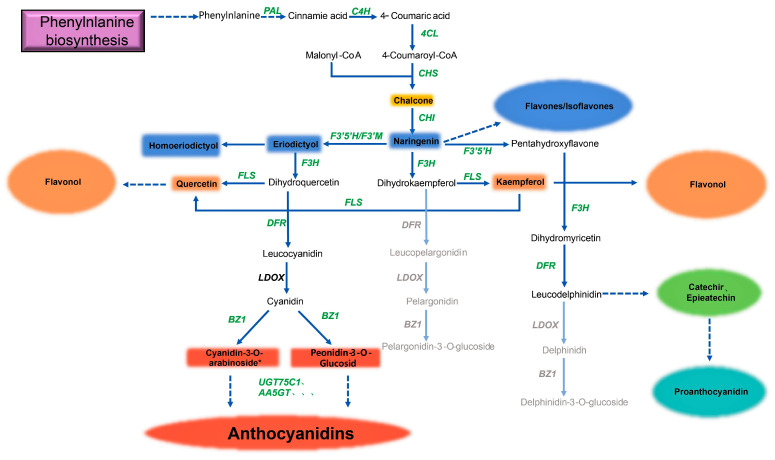
Remodeling of the flavonoid metabolism pathway in *Deinococcus* sp. 43. Genes annotated in the *Deinococcus* sp. 43 genome are shown in bold green text; genes unannotated in the *Deinococcus* sp. 43 genome are shown in bold black text; the flavonoids identified in the metabolite detection are shown in colored boxes; the quadrangular box represents a specific substance, while the circular box represents a class of flavonoids. Different colored boxes represent different flavonoids. Catalytic enzyme genes of the flavonoid metabolism pathway are present.

**Table 1 microorganisms-11-01848-t001:** PCR primers for RT-qPCR analysis.

Gene	Gene	Sequence of Primers (5′–3′)
PAL-F	*PAL*	AACCCTCTGATCTTCCCCGA
PAL-R		ACTTTCAGCGTGTCGATGGT
CHS-F	* CHS *	ACATGAGTAGCGTCACCGTC
CHS-R		AGTTCAGGAGGACGTGTTCG
F3H-F	* F3H *	GGAACGGGATCGAGGTCTAC
F3H-R		CACGGCTTTCGTGTCCATCT
CHI-F	* CHI *	GGCGTCGGTCGCCTATTT
CHI-R		TCAGCCACGCGTGATTCAG
RLPL32-F	* RLPL *	GTCCCCAAGAAGAAGACCAGC
RLPL32-R		CCTGGCGGCCATCGTAGTAA

**Table 2 microorganisms-11-01848-t002:** Statistics of gene assembly and function annotation.

Attributes	Characteristic
Clean reads	4,017,618
Clean base	1,205,285,400
Q20 (%)	96.28%
N50	203,832 bp
Total base	4,314,008 bp
GC content (%)	69.53%
CDS	4116
tRNA genes	56
rRNA genes	2
Genes assigned to NR	4036 (96.67%)
Genes assigned to COG	3252 (77.89%)
Genes assigned to KEGG	1808 (43.31%)

**Table 3 microorganisms-11-01848-t003:** The intensity of flavonoids in *Deinococcus* sp. 43 intracellular (DI) and extracellular (DE) environments (the flavonoids already present in the medium were excluded). *Deinococcus* sp. 43 produces 56 kinds of flavonoids, and the flavonoids of intracellular and extracellular environments show significant differences (The symbol * represents an unclear configuration of the substance).

Compound	DI	DE	*p* Value
Hesperetin	1.28 × 10^4^	6.99 × 10^4^	0.001178
Quercetin	2.62 × 10^4^	2.77 × 10^4^	0.238109
7-O-Methylnaringenin	0	1.76 × 10^4^	0.004591
3,4′-Dihydroxyflavone	3.16 × 10^5^	1.52 × 10^6^	0.001678
Kaempferol-3-O-(6″-Acetyl) glucosyl -(1→3)-Galactoside	2.22 × 10^5^	6.75 × 10^3^	6.36 × 10^−5^
7-O-Methyleriodictyol	0	4.63 × 10^4^	0.033842
Tricin (5,7,4′-Trihydroxy-3′,5′-dimethoxyflavone)	0	3.52 × 10^5^	0.012179
5,7,2′-Trhiyroxy-8-methoxy flavone	0	5.00 × 10^5^	0.008064
Naringenin (5,7,4′-Trihydroxyflavanone)	0	1.44 × 10^5^	0.03777
Tricin-4′-O-(guaiacylglycerol) ether	0	6.62 × 10^5^	0.008651
Kaempferol-3-O-rutinoside-7-O-rhamnoside	3.15 × 10^3^	0	0.000327
Gnetifolin B	0	9.34 × 10^4^	0.009861
Chrysoeriol; 5,7,4′-Trihydroxy-3′-Methoxyflavone	0	1.48 × 10^5^	0.004042
Naringenin chalcone; 2′,4,4′,6′-Tetrahydroxychalcone	0	1.55 × 10^5^	0.039474
Chrysoeriol-7-O-homovanillic acid	0	1.55 × 10^4^	0.004311
Tricin-4′-O-syringic acid	0	2.18 × 10^4^	0.110615
Tricin-4′-O-oxalic acid-7-O-(p-coumaroyl) shikimic acid	0	3.39 × 10^4^	0.141641
Kaempferol-3,7-O-diglucoside	1.80 × 10^4^	0	0.035479
Tricin-4′-O-eudesmic acid	0	3.17 × 10^4^	0.040095
Apigenin; 4′,5,7-Trihydroxyflavone	0	1.90 × 10^4^	0.015337
Psoralenol	4.76 × 10^4^	8.14 × 10^4^	0.003094
Tricin-7-O-(2″-O-rhamnosyl) galacturonide	3.72 × 10^5^	0	0.147061
Kaempferol-3-(2″,6″-di-O-rhamnosyl)-glucoside	4.58 × 10^4^	0	0.000393
Cyanidin-3-O-arabinoside *	4.78 × 10^4^	0	0.005653
Eriodictyol (5,7,3′,4′-Tetrahydroxyflavanone)	0	3.89 × 10^4^	0.016676
Apigenin-6-C-glucoside (Isovitexin)	0	1.82 × 10^5^	0.017304
Quercetagetin; 3,3′,4′,5,6,7-Hexahydroxyflavone	0	5.37 × 10^4^	0.003139
Tricin-4′-O-glycerol	0	3.79 × 10^4^	0.011644
Tricin-7-O-(2″-Malonyl) rhamnoside	0	3.72 × 10^4^	0.146934
Isorhamnetin-3-O-rutinoside-7-O-rutinoside	6.72 × 10^4^	0	0.000119
Apigenin-8-C-Glucoside (Vitexin)	0	1.01 × 10^5^	0.026334
Tricin-4′-O-syringyl alcohol	0	7.86 × 10^4^	0.003272
Kaempferol (3,5,7,4′-Tetrahydroxyflavone)	0	7.13 × 10^3^	0.051729
Malvidin-3,5-di-O-glucoside	4.73 × 10^3^	0	0.005893
Chrysoeriol-7-O-(6″-acetyl) glucoside	0	1.76 × 10^3^	0.017605
Tricin-7-O-rutinoside *	3.96 × 10^3^	3.02 × 10^5^	0.039741
Gallocatechin-(4α→8)-gallocatechin	9.73 × 10^4^	1.30 × 10^3^	0.037396
Apigenin-7-O-neohesperidoside (Rhoifolin) *	0	1.42 × 10^5^	0.007817
Kaempferol-3-O-(2-O-Xylosyl-6-O-Rhamnosyl) Glucoside	2.00 × 10^4^	1.41 × 10^3^	0.026179
Chrysoeriol-7-O-glucoside	1.77 × 10^4^	0	0.020361
Mirificin	3.05 × 10^4^	0	0.004285
Chrysoeriol-5,7-di-O-rutinoside	1.52 × 10^4^	0	0.014933
Homoeriodictyol	0	1.15 × 10^4^	0.004614
Chrysoeriol-6,8-di-C-glucoside	1.87 × 10^5^	0	0.002926
Apigenin-7-O-(6″-p-Coumaryl) glucoside	0	1.45 × 10^5^	0.019662
Nepetin (5,7,3′,4′-Tetrahydroxy-6-methoxyflavone)	0	1.64 × 10^4^	0.002265
Procyanidin B1	0	1.53 × 10^3^	0.000211
Quercetin-5,7,4′-tri-O-glucoside	2.36 × 10^4^	0	0.002187
Tricin-7-O-neohesperidoside *	0	2.06 × 10^5^	1.21 × 10^−6^
Luteolin-7-O-gentiobioside	7.24 × 10^4^	1.42 × 10^3^	0.03487
Apiin	3.06 × 10^4^	3.42 × 10^4^	0.285151
Eriodictyol-3′-O-glucoside	0	1.49 × 10^4^	0.027668
Apigenin-7-O-rutinoside (Isorhoifolin) *	0	1.30 × 10^5^	1.61 × 10^−5^
Dihydrokaempferol-7-O-glucoside	0	1.45 × 10^4^	0.012045
Catechin gallate	1.71 × 10^4^	0	0.011896
kaempferol-3-O-α-rhamnopyranosyl-(1→2)- [α-rhamnopyranosyl-(1→6)]-β-galactopyranoside-7-O-α-rhamnopyranoside	4.48 × 10^4^	0	0.002381

## Data Availability

The datasets presented in this study can be viewed in online repositories. The names of the repository(s) and accession numbers can be found at https://www.ncbi.nlm.nih.gov/bioproject/, accessed on 17 November 2022. PRJNA902113. In BioProject PRJNA902113, the assembled genome of *Deinococcus* sp. 43 is available with the accession number SAMN31747608. A filtered strain has been submitted to the CCTCC, Wuhan University, China, with the accession ID in the CCTCC NO: M 2022653.

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
