# Peer review of "Flavonoid Synthesis by *Deinococcus* sp. 43 Isolated from the *Ginkgo* Rhizosphere"

_microorganisms, 2023, doi:10.3390/microorganisms11071848_

Round 1
Reviewer 1 Report
In this study, Deinococcus sp. 43, a flavonoid-producing bacteria from the Ginkgo rhizosphere was reported. The Authors performed genomic analysis of Deinococcus sp. 43, demonstrated its flavonoid-producing capacity and the expression of key genes involved in flavonoid synthesis. A total of 76 flavonoid compounds have been identified. Moreover, on the basis of genomic analysis and the detected flavonoids, the flavonoid biosynthetic pathway of Deinococcus sp. 43 was reconstructed for the first time. The manuscript provides significant insights into the flavonoid-producing potential of the studied bacterial strain.
The manuscript is well-structured. The introduction section provides sufficient background on the topic. The methods are described in detail. The results were presented systematically, and the discussion is comprehensive. The conclusions are consistent with the obtained results. However, the quality of English language needs improvements.
The following should be addressed:
- The title of the manuscript should be corrected – instead of “Flavonoid synthetic” use “Flavonoid synthesis”.
- Was the species collected from a high-radiation environment? Which environmental factors, other than radiation could influence flavonoid secretion by Deinococcus?
- In line 309, 310 tricin was repeated twice one after another
- Species names should be written in italic (lines 420, 426)
The quality of English language needs improvement. It would benefit from proofreading by a native speaker.
For example, the second sentence in the abstract should be improved for grammar mistakes: FLavonoids are crucial; instead of starting the sentence with "And" (And the relative expression of key genes...) it would be more appropriate to use "Moreover or Furthermore". Also, some sentences in Material and Methods, such as in lines 141-144 should be rephrased or split into several shorter.
Author Response
Point1: For example, the second sentence in the abstract should be improved for grammar mistakes: Flavonoids are crucial; instead of starting the sentence with "And" (And the relative expression of key genes...) it would be more appropriate to use "Moreover or Furthermore". Also, some sentences in Material and Methods, such as in lines 141-144 should be rephrased or split into several shorter.
Response 1: Thank you for your valuable feedback. The format and grammar of the article have been revised according to your suggestions. ( All the revisions in the revised manuscript are highlighted in yellow.)
Point 2: The title of the manuscript should be corrected – instead of “Flavonoid synthetic” use “Flavonoid synthesis”.
Response 2: We have changed the "Flavonoid synthetic" in the title to "flavonoid synthesis" as you suggested(line1~line2).
Point 3: Was the species collected from a high-radiation environment? Which environmental factors, other than radiation could influence flavonoid secretion by Deinococcus?
Response 3: We are extremely grateful to Reviewer for pointing out this problem. The Deinococcus sp .43 in this study is isolated from the rhizosphere soil of Ginkgo. Further experiments are needed to investigate the effects of specific environmental factors on flavonoid secretion.
Point 4: In line 309, 310 tricin was repeated twice one after another.
Response 4: The redundant portion has been eliminated(line311~line312).
Point 5: Species names should be written in italic (lines 420, 426)
Response 5: We have changed the species names to italics (line426~line427).

Reviewer 2 Report
on line 109 add the geographic coordinates of the location of the sampling site
Author Response
Point 1: On line 109 add the geographic coordinates of the location of the sampling site.
Response 1: Thank you for your valuable feedback. We are extremely grateful to Reviewer for pointing out this problem. Geographic data have been added to the experimental method(line107). At the same time, the format and grammar of the article have been revised.( All the revisions in the revised manuscript are highlighted in yellow.)

Reviewer 3 Report
Kindly refer to the attached document.

Kindly refer to the comments and suggestions attached for the authors.
Author Response
Point 1: The results of this study are noteworthy and deserve to be published. However, numerous grammatical mistakes in the English language need to be addressed. For example, all methods should be written in the past tense.
Response 1: Thank you for your valuable feedback. The format and grammar of the article have been revised according to your suggestions.( All the revisions in the revised manuscript are highlighted in yellow.)
Point 2: Can the authors explain why the concentration of certain glycosylated flavonoids is significantly higher outside the cell than inside, but this is not the case for all glycosylated flavonoids mentioned in Table 3?
Response 2: We are extremely grateful to Reviewer for pointing out this problem. By examining the flavonoids with the most pronounced differences in content inside and outside the cell, we found that the 20 compounds with the largest differences showed this pattern, and the substances with the largest differences were more abundant. We have simply explained the rules of these data, we have also hypothesized some potential causes for this phenomenon. At the same time, there may be some glycosylated flavonoids in the cell. This may be because glycosylation is a cell-based modification that must cross the cell membrane to reach the outside of the cell, which leads to the detection of some glycosylated flavonoids in the cell.
Point 3: The description below Figure 2 should be revised in order to provide a better understanding of the figures' content.
Response 3: Thank you for your valuable feedback. We have modified the description of numbers below Figure 2(line271~line282):
Figure 2. At various TGY concentrations and fermentation durations, it displays the growth of bacteria and the generation of quercetin. a demonstrates how the concentration of the medium has a significant impact on the yield of quercetin. In the medium containing 45 g/L, the highest quercetin yield (1.9 mg/L) was attained. When the medium concentration was less than 25g/L, quercetin was hardly produced by Deinococcus sp. 43. On the whole, the yield of quercetin is positively correlated with the medium concentration. b shows the change of quercetin in fermentation broth at different times. The quercetin production of Deinococcus sp. 43 generally showed a trend of first rising and then decreasing. Quercetin production reached its maximum at 72 hours. In response to variations in flavonoid production, both c and d demonstrate the relative expression of some essential genes involved in flavonoid synthesis. We can find that the expression of flavonoid synthesis genes increases with increased flavonoid production. In particular, under different medium concentrations, the expression of flavonoid synthesis genes varied tens of times.
Point 4: Please reformat the references in MDIP format.
Response 4: Thank you for your valuable feedback. We have changed the format of the reference to MDPI format(line539~line698).
